# A Keplerian Ag$_{90}$ nest of Platonic and Archimedean polyhedra in different symmetry groups

Yan-Min Su [1,4], Zhi Wang[1,4], Stan Schein [2✉], Chen-Ho Tung[1] & Di Sun [1,3✉]

Polyhedra are ubiquitous in chemistry, biology, mathematics and other disciplines. Coordination-driven self-assembly has created molecules mimicking Platonic, Archimedean and even Goldberg polyhedra, however, nesting multiple polyhedra in one cluster is challenging, not only for synthesis but also for determining the alignment of the polyhedra. Here, we synthesize a nested Ag$_{90}$ nanocluster under solvothermal condition. This *pseudo*-T$_h$ symmetric Ag$_{90}$ ball contains three concentric Ag polyhedra with apparently incompatible symmetry. Specifically, the inner (Ag$_6$) and middle (Ag$_{24}$) shells are octahedral (O$_h$), an octahedron (a Platonic solid with six 3.3.3.3 vertices) and a truncated octahedron (an Archimedean solid with twenty-four 4.6.6 vertices), whereas the outer (Ag$_{60}$) shell is icosahedral (I$_h$), a rhombicosidodecahedron (an Archimedean solid with sixty 3.4.5.4 vertices). The Ag$_{90}$ nanocluster solves the apparent incompatibility with the most symmetric arrangement of 2- and 3-fold rotational axes, similar to the arrangement in the model called Kepler's Kosmos, devised by the mathematician John Conway.

[1] Key Laboratory of Colloid and Interface Chemistry, Ministry of Education, School of Chemistry and Chemical Engineering, State Key Laboratory of Crystal Materials, Shandong University, Ji'nan 250100, People's Republic of China. [2] California NanoSystems Institute and Department of Psychology, University of California, Los Angeles, CA 90095-1563, USA. [3] Shandong Provincial Key Laboratory of Chemical Energy Storage and Novel Cell Technology, and School of Chemistry and Chemical Engineering, Liaocheng University, Liaocheng 252000, People's Republic of China. [4] These authors contributed equally: Yan-Min Su, Zhi Wang. ✉email: stan.schein@gmail.com; dsun@sdu.edu.cn

Encouraged by masterpieces of self-assembly in biology[1–5], some seminal metal clusters[6] of nanometer size have been assembled from simple components[7–12], such as $Al_{77}$[13], $Gd_{140}$[14], $Pd_{145}$[15], $Ag_{374}$[16], and $Au_{246}$[17]. Among families of metal clusters, silver nanoclusters benefit from the exceptional versatility of silver(I) atoms, which have flexible coordination preferences, a tendency to form argentophilic interactions and a susceptibility to reduction, all properties enriching the number of members and types of this family[18–20]. To obtain structures with polyhedral geometry[21–24], silver polygons can form with the assistance of surface ligands, inner anion templates and argentophilic interactions. Nonetheless, most silver nanoclusters lack typical polyhedral features. Exceptionally, in 2017, we synthesized a buckyball-like Goldberg cage with 180 Ag atoms[25].

Even more complex is a nested silver(I) nanocluster with two or more chemically bound metallic shells. Examples include $Ag_{56}$ ($Ag_{14} \subset Ag_{42}$)[26], $Ag_{60}$ ($Ag_{12} \subset Ag_{48}$)[27], $Ag_{62}$ ($Ag_{14} \subset Ag_{48}$)[28], and $Ag_{78}$ ($Ag_{18} \subset Ag_{60}$)[29], but even in these cases, most of the shells lack typical polyhedral features, and the number of silver shells is just two. We assembled the first three-shell $Ag_{73}$ silver nanocluster with a central Ag atom in an $Ag_{24}$ rhombicuboctahedron in an $Ag_{48}$ octahedral Goldberg 2,0 polyhedron[30]. Although there has been progress, the synthetic challenges in making single polyhedral and nested polyhedral silver nanoclusters remain due to the difficulty of precisely shaping silver polygons and obtaining polyhedral shells with compatible symmetry. Here, we report the synthesis and characterization of nested $Ag_{90}$ nanoclusters which show *pseudo*-$T_h$ symmetry and contain three concentric silver polyhedra with apparently incompatible symmetry. The nested silver shells can be expressed as $Ag_6 \subset Ag_{24} \subset Ag_{60}$ and belong to octahedron, truncated octahedron, and rhombicosidodecahedron, respectively. The $Ag_{90}$ nanocluster solves the apparent incompatibility with the most symmetric arrangement of two- and threefold rotational axes.

## Results

**Structures of SD/Ag90a and SD/Ag90b.** With a combination of anion-template and geometrical-polyhedron strategies[31–37] and with careful choice of organic ligands ($^tBuSH$ and $PhPO_3H_2$) and anion templates ($S^{2-}$ and $PO_4^{3-}$), we one-pot synthesized $Ag_{90}$ nanoclusters under the solvothermal condition as dark brown or red rhombic crystals, depending on the polymorphs. They are stable under ambient conditions because the bulk sample of them can keep the color and morphology unchanged for at least one month. The $PO_4^{3-}$ ion was considered as anion-template mainly due to its high negative charge and $T_d$ symmetry. The former feature can aggregate more $Ag^+$ ions to form high-nuclearity cluster through electrostatic attraction, whereas the latter can shape cluster with a T-related symmetry. The $Ag_{90}$ nanoclusters can be crystallized into monoclinic $P2_1/n$ or triclinic $P$-1 phases dictated by the silver salts used (Fig. 1), hereafter denoted as **SD/Ag90a** and **SD/Ag90b**, respectively. Although the different anions, $PhCOO^-$ and $CH_3SO_3^-$, did not participate in the final structures of the $Ag_{90}$ clusters, they may have influenced the crystallization process through supramolecular interactions such as hydrogen bonds, which causes the formation of the ultimate crystalline phases. Other common silver salts, such as $AgBF_4$, $CF_3COOAg$, $CF_3SO_3Ag$, and $AgNO_3$, were tried in the above synthesis experiments, but none produced desired clusters. A series of characterization techniques such as single-crystal X-ray crystallography, infrared spectroscopy, $^{31}P$ nuclear magnetic resonance, ultraviolet–visible absorption spectroscopy, fluorescence spectroscopy, and thermogravimetric analysis were used on these two nests (Supplementary Figs. 1–5, 13–17 and Supplementary Tables 1–3). The electrospray ionization mass

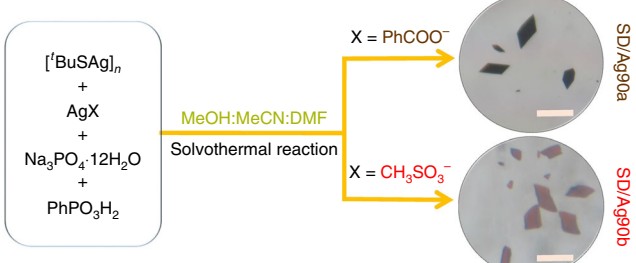

**Fig. 1 Synthetic routes for SD/Ag90a and SD/Ag90b.** The photos of crystals were taken in the ambient environment with a digital camera. X represents the counter-anions in the silver salts (AgX) used in the syntheses. DMF = *N,N*-dimethylformamide. Scale bar: 0.3 mm.

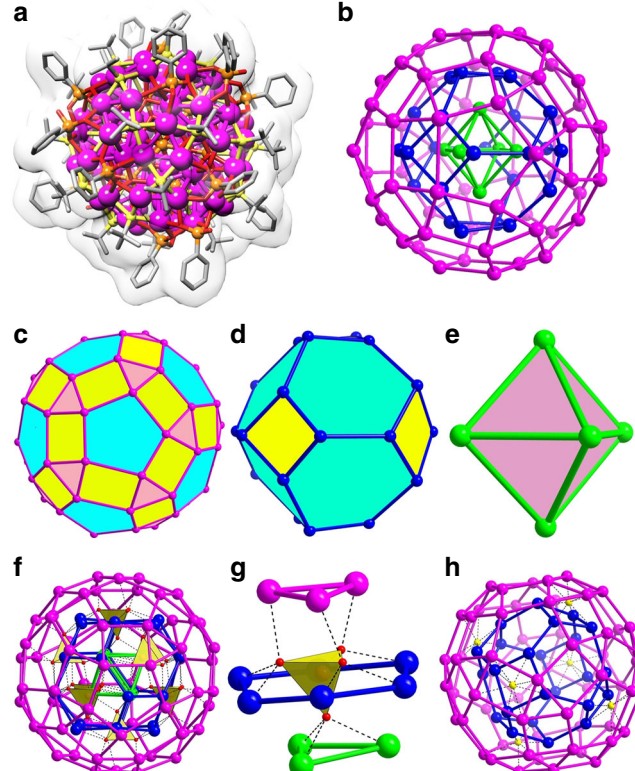

**Fig. 2 Single-crystal X-ray structure of SD/Ag90a. a** Total cluster structure of **SD/Ag90a** incorporating Van der Waals surfaces. Hydrogen atoms are removed for clarity. Color legend: Ag, purple; P, brown; S, yellow; O, red; C, gray. **b** The ball-and-stick mode of the triply nested polyhedral silver skeleton, viewed down an axis through the front silver 5-gon. The three different shells are individually colored. **c** The icosahedral $Ag_{60}$ rhombicosidodecahedron. **d** The octahedral $Ag_{24}$ truncated octahedron. **e** The octahedral $Ag_6$ octahedron. **f** The interactions between eight $PO_4^{3-}$ and three different shells. **g** The detailed coordination of $PO_4^{3-}$ towards different silver polygons in three different shells. All $PO_4^{3-}$ ions are shown as yellow tetrahedra. **h** Six $\mu_8$-$S^{2-}$ ions intercalate the aperture between $Ag_{24}$ and $Ag_{60}$ shells of **SD/Ag90a** by linking two 4-gons up and down from these shells, respectively.

spectrometry (ESI-MS) of **SD/Ag90a** dissolved in $CH_2Cl_2$ or $CH_3OH$ did not give useful data, which indicates that either (i) **SD/Ag90a** is fragmented during the ionization process or (ii) it is neutral and is hard to ionize under mass spectrometer conditions —even when we added CsOAc to aid in ionization[38].

As deduced from crystallography, **SD/Ag90a** (Fig. 2a) and **SD/Ag90b** (Supplementary Fig. 6a) have the same molecular

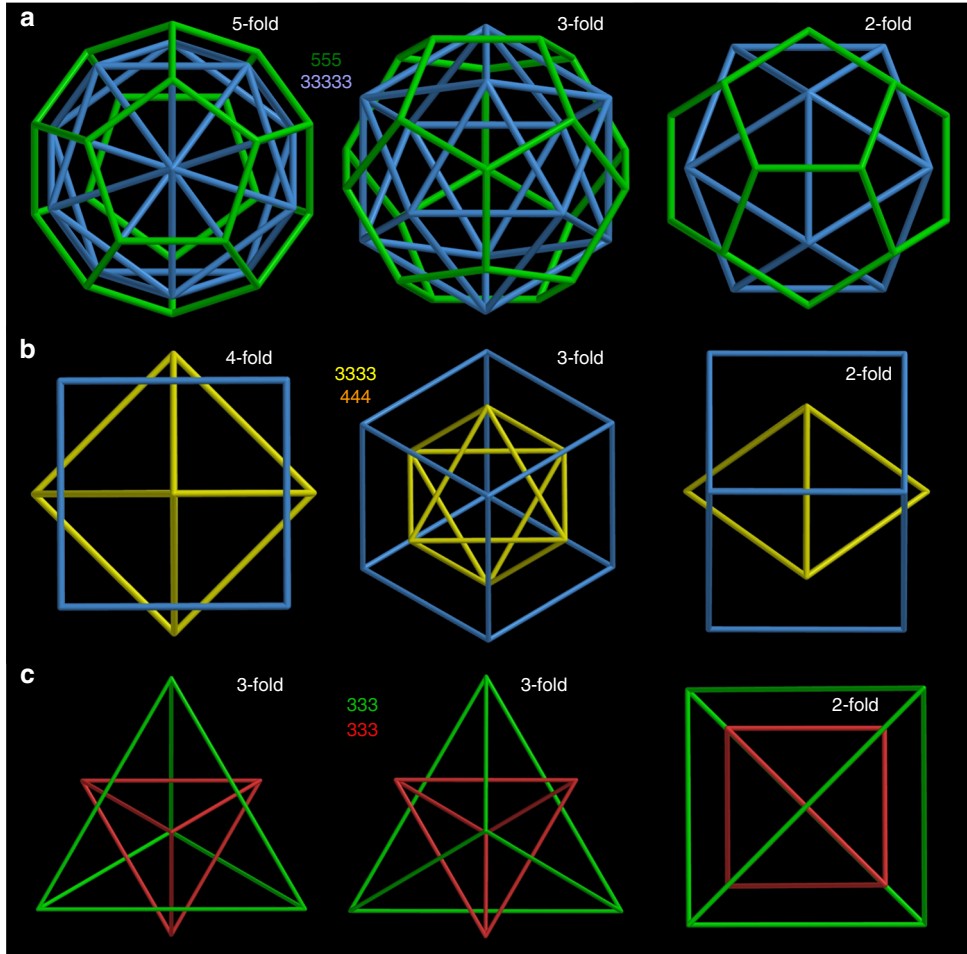

**Fig. 3 Alignment of pairs of icosahedral, octahedral, and tetrahedral Platonic solids.** The solids, arbitrarily scaled, are labeled by their vertex descriptions: the icosahedral dodecahedron (555) and icosahedron (33333), the octahedral cube (444) and octahedron (3333), and the tetrahedral tetrahedron (333), where 5, 4, and 3 represent faces, respectively, 5-gons, 4-gons, and 3-gons. **a** Arrangement of the icosahedral solids with alignment along five-, three- and twofold rotational axes. **b** Arrangement of the octahedral solids with alignment along four-, three- and twofold axes. **c** Arrangement of the tetrahedron with its self-dual, another tetrahedron, with alignment along threefold axes with a face in front, threefold axes with a vertex in front, and twofold axes.

structure and composition of $[(PO_4)_8@Ag_{90}S_6(^tBuS)_{24}(Ph-PO_3)_{12}(PhPO_3H)_6]$, but they crystallize into different space groups (Supplementary Table 1) due to different cluster packing in their unit cells (Supplementary Figs. 7 and 8). The straightforward single-crystal X-ray diffraction (SCXRD) of crystals at 100 K unambiguously established ball-shaped structures for the $Ag_{90}$ nanoclusters (Fig. 2b–e; Supplementary Figs. 6b–e, 9, 10). Due to the similarity of their molecular structures, we take **SD/Ag90a** as representative for discussions in detail below.

**SD/Ag90a** crystallized into the monoclinic $P2_1/n$ space group and conformed to *pseudo*-$T_h$ symmetry. **SD/Ag90a** is a neutral cluster with all Ag(I) atoms in 3- or 4-coordination with S and/or O atoms. The all-silver framework (Fig. 2b) is composed of three concentric nested polyhedra, an outer $Ag_{60}$ rhombicosidodecahedron with 60 3.4.5.4 vertices (Fig. 2c), a middle $Ag_{24}$ truncated octahedron with 24 4.6.6 vertices (Fig. 2d), and an inner $Ag_6$ octahedron with 6 3.3.3.3 vertices (Fig. 2e), where the numbers 5, 4, and 3 represent faces around that vertex, respectively, 5-gons, 4-gons, and 3-gons. Of note, the outer $Ag_{60}$ shell is geometrically reminiscent of the third shell ($Pd_{60}$) in Dahl's $Pd_{145}$ cluster[15]. All vertices on these three polyhedra are Ag(I) atoms, and all edges are built from the connection of adjacent two Ag atoms. The Ag–Ag edge lengths in outer, middle, and inner shells range from 2.96 to 4.03, 3.02 to 3.48, and 3.51–3.61 Å, respectively

(Supplementary Fig. 11 and Supplementary Table 3). Some of these Ag···Ag edge lengths, shorter than 3.44 Å, twice the Van der Waals radius of silver(I) ion, can be deemed as argentophilic interactions that contribute to the stability of the silver shells. Of note, the long Ag···Ag edges in the $Ag_6$ octahedron also rule out a subvalent nature, which usually produces short Ag···Ag distances approximating to 2.88 Å[39]. By measuring distances between inversion-related pairs of Ag atoms in the same shell, the diameters of outer, middle and inner shells are determined to be 1.5, 1.0, and 0.5 nm, respectively.

Each $\mu_{12}$-$\kappa^3$:$\kappa^3$:$\kappa^3$:$\kappa^3$ $PO_4^{3-}$ ion (as anion-template) penetrates the hexagonal windows of the $Ag_{24}$ shell to connect all three silver shells (Ag-O: 2.3–2.6 Å) by linking two silver 3-gons of the $Ag_6$ and $Ag_{60}$ shells and one 6-gon of the $Ag_{24}$ shell (Fig. 2f, g). Six $\mu_8$-$S^{2-}$ ions from in situ decomposition of $^tBuSH$ intercalate the aperture between $Ag_{24}$ and $Ag_{60}$ shells (Ag-S: 2.43–2.89 Å) by linking two 4-gons up and down from these shells, respectively (Fig. 2h)[28]. Based on the above analysis, we found that the tetrahedral $PO_4^{3-}$ ion has a special role in shaping silver 3-gons and 6-gons, essential elements to construct the rhombicosidodecahedron and the truncated octahedron, respectively. As for the spherical $S^{2-}$ ion, it assists in fabricating the silver 4-gon, an essential element for both the rhombicosidodecahedron and the truncated octahedron. Both inorganic anions act not only as templates to shape the silver polyhedra by defining the essential

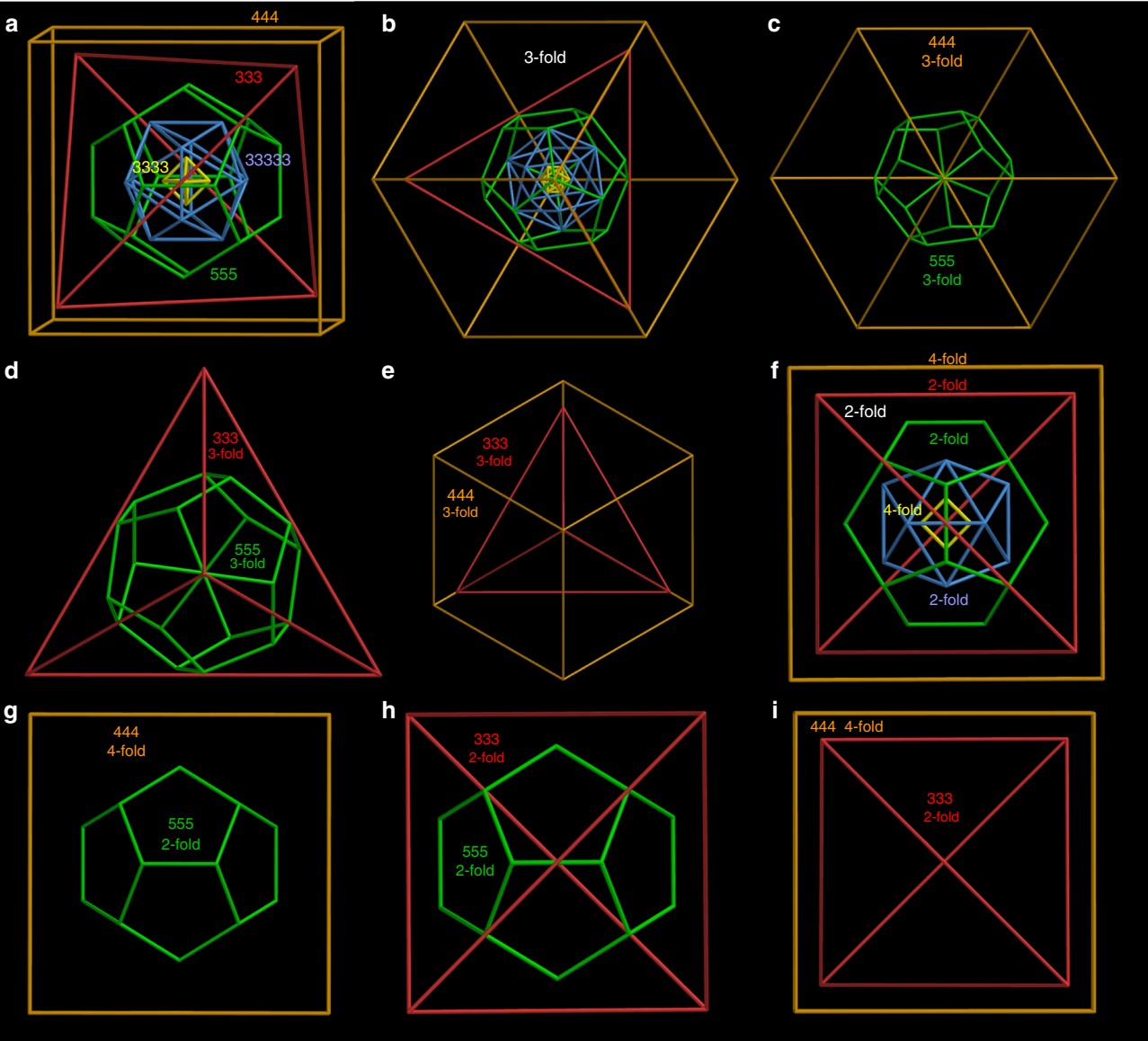

**Fig. 4 T_d arrangement of the five Platonic solids in the "Keplers Kosmos" model. a** A view of the quintuple nest, slightly off a twofold axis, and **b** along a common threefold axis. **c** Views along the threefold axis of the double 555 and 444 nest, **d** the double 555 and 333 nest, and **e** the double 444 and 333 nest. **f** A view of the quintuple nest along four- and twofold axes. **g** Views of the double 555 and 444 nest along two- and fourfold axes, **h** the double 555 and 333 nest along twofold axes, and **i** the double 444 and 333 nest along four- and twofold axes. The solids are arbitrarily scaled.

polygon elements but also function as glue to consolidate the overall nested silver shells.

The ligand coverage on the surface of the outer Ag$_{60}$ shell is polygon selective with 24 $^t$BuS$^-$ and 6 PhPO$_3$H$^-$ on thirty 4-gons and 12 PhPO$_3^{2-}$ on twelve 5-gons. There are no ligands capping the twenty 3-gons. The PhPO$_3$H$_2$ ligand exhibits two kinds of deprotonated forms, PhPO$_3^{2-}$ and PhPO$_3$H$^-$, that respectively coordinate with twelve 5-gons ($\mu_5$-$\kappa^2$:$\kappa^2$:$\kappa^1$) and six 4-gons ($\mu_4$-$\kappa^2$: $\kappa^2$) on the surface of the Ag$_{60}$ shell (Ag-O: 2.2–2.6 Å). The $^{31}$P NMR (nuclear magnetic resonance) of the digestion solution of **SD/Ag90a** shows two sharp peaks with chemical shifts at $\delta =$ −1.07 and 15.62 ppm (Supplementary Fig. 2), corresponding to H$_3$PO$_4$ and PhPO$_3$H$_2$, respectively, which clearly verify the existence of two different P-containing chemicals in **SD/Ag90a**. From the ligation modes of each coordinative component, we suggest that all of them play roles to shape different silver polygons, paving the way for further construction of polyhedral silver nanoclusters. The overall structure is reinforced by a

combination of argentophilic interactions (<3.44 Å) and the scaffolding provided by all other coordination bonds. The remarkable structure of **SD/Ag90a** has not been previously observed in the family of silver nanoclusters.

**Alignment of shells with compatible point-group symmetry.** We now ask about the alignment of the icosahedral and octahedral cages in the **SD/Ag90a** nest. For the dodecahedron and its dual (the icosahedron), both Platonic solids with icosahedral (I$_h$) symmetry and thus "compatible", nesting may be based on alignment of all of the five-, three- and twofold axes of rotational symmetry (Fig. 3a; Supplementary Table 4). The same full alignment could obtain for nests of any icosahedral structures, including the six icosahedral Archimedean solids and an infinity of other icosahedral structures. Likewise, nesting of octahedral shells like the cube and its dual (the octahedron)—both Platonic solids—with each other (Fig. 3b) and of tetrahedral shells like the

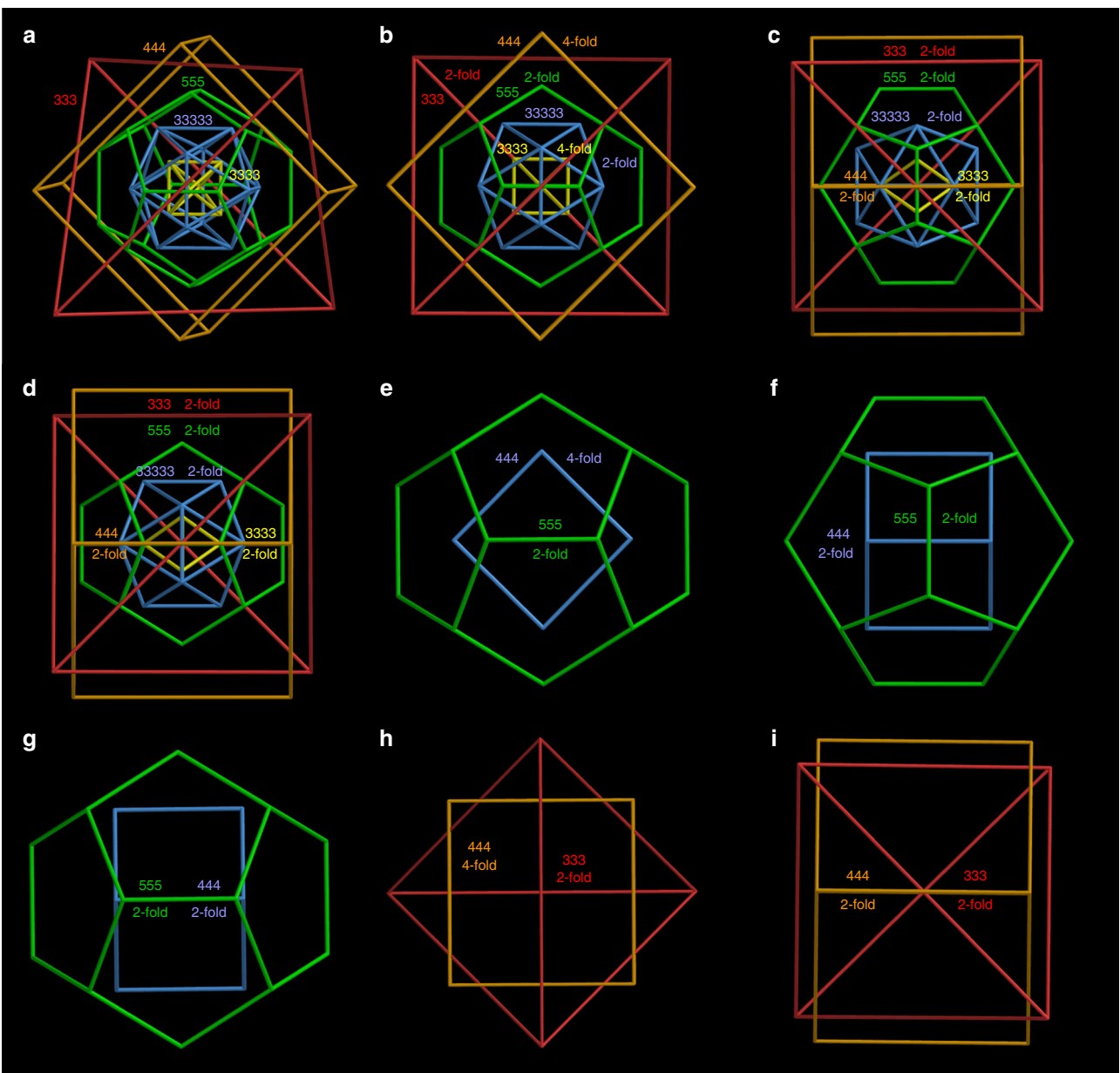

**Fig. 5 An alternative (D₂) arrangement of the five Platonic solids. a** Views of the quintuple nest, arbitrarily scaled, slightly off a twofold axis, **b** along four- and twofold axes, **c** along twofold axes (with edges at right angles), and **d** along twofold axes (with edges parallel). **e** Views of the double 555 and 444 nest along two- and fourfold axes, **f** along twofold axes (with edges at right angles, corresponding to **c**), and **g** along twofold axes (with edges parallel, corresponding to **d**). **h** Views of the double 444 and 333 nest along four- and twofold axes and **i** along twofold axes. With just three (orthogonal) twofold axes, each one different from the other, but with no mirrors, this regular quintuple nest has D₂ symmetry.

tetrahedron and its self-dual (the tetrahedron) (Fig. 3c) with each other may be based on alignment of all rotational axes (Supplementary Table 4). Indeed, the Ag₇₃ cited above is just such a symmetry-compatible nesting of octahedral silver cages[30].

**Alignment of shells with incompatible point-group symmetry.** Although it might be assumed that only cages with compatible symmetry (e.g., icosahedral with icosahedral) could be nested, for the Zometool toy the mathematician John Conway created a model called "Kepler's Kosmos", a model that aligns the five Platonic solids, the two with icosahedral symmetry, the two with octahedral, and the one with tetrahedral[40]. As its name suggests, the inspiration for this model dates back to Johannes Kepler. In

1596, in his Mysterium Cosmographicum (The Secret of the Universe)[41], Kepler hypothesized that the orbits of the six known planets corresponded with six spheres, five circumscribing the five Platonic solids and one inscribing the smallest. To test his hypothesis, Kepler became mathematical assistant to Tycho Brahe in 1600 and gained access to more than 30 years of astronomical observations. By 1605, Kepler had shown that the orbit of Mars was an ellipse, culminating by 1619 with his discovery of the three laws of planetary motion[42, 43] and subsequently Isaac Newton's discovery of the law of universal gravity[44].

Kepler's Kosmos provides one possible answer to the question of how to align icosahedral polyhedra with fivefold rotational axes but no fourfold, octahedral polyhedra with fourfold axes but no fivefold, and tetrahedral polyhedra with neither. Conway's

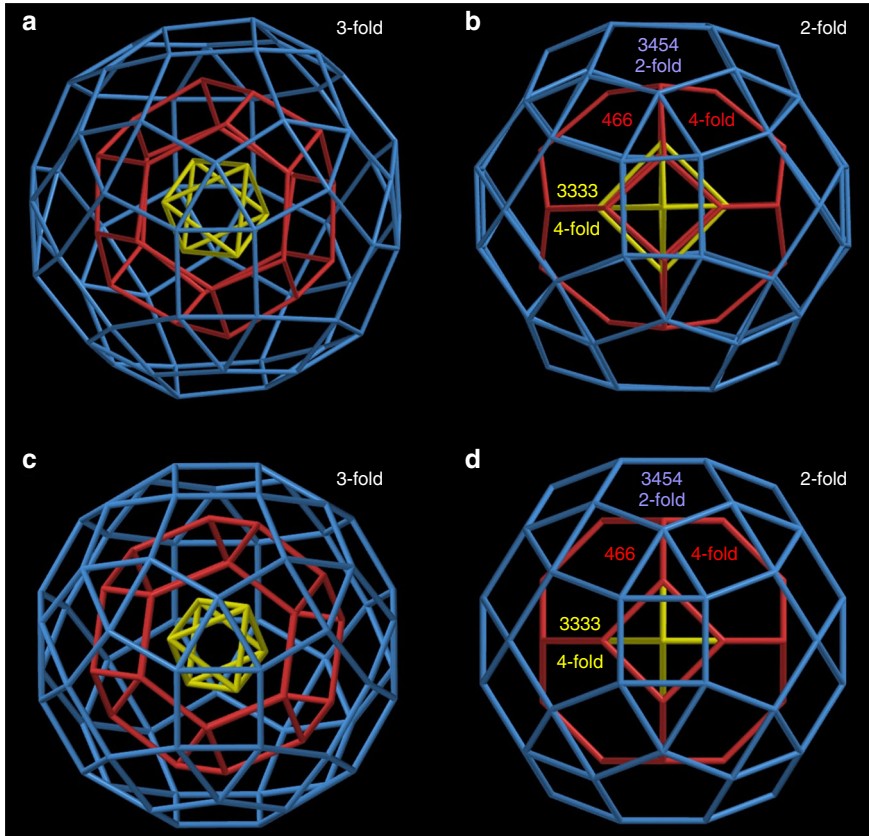

**Fig. 6 Silver shells of SD/Ag90a. a** A view through the common threefold axes of **SD/Ag90a**. There are four of these rotational axes. **b** A view through a twofold axis of the outer icosahedral Ag shell of **SD/Ag90a**, which corresponds with a fourfold axis through the middle and inner octahedral Ag shells. There are three of these rotational axes, arranged orthogonally. **c, d** The corresponding triple nest with regular polyhedral shells viewed along the three- and twofold axes of the nest.

arrangement (Fig. 4a; Supplementary Fig. 12a and Supplementary Table 4) has these properties: (i) None of the six fivefold axes of the dodecahedron (or icosahedron) is aligned with any rotational axis of an octahedral or tetrahedral polyhedron. (ii) Four of the ten threefold axes of the dodecahedron (or icosahedron) are aligned with all four threefold axes of the cube (or octahedron) and all four threefold axes of the tetrahedron (Fig. 4b–e). (iii) Three of the fifteen twofold axes of the dodecahedron (or icosahedron) are aligned with all three (orthogonal) fourfold axes of the cube (or octahedron) and all three (orthogonal) twofold axes of the tetrahedron (Fig. 4f–i). This quintuple nest itself has tetrahedral ($T_d$) symmetry.

However, other alignments are possible. Here, we devise an alternative alignment with three different combinations of four- and twofold axes to produce a nest with just three different, orthogonal, twofold axes (Fig. 5; Supplementary Fig. 12b, Supplementary Table 4, and Supplementary Movie 1), producing a quintuple nest with lower ($D_2$) symmetry. Of course, it is also possible to align none of the rotational axes, producing a quintuple nest with trivial ($C_1$) symmetry.

**Alignment of shells in SD/Ag90a.** Given different symmetric arrangements of the three shells—as in the Kepler's Kosmos with point-group $T_d$ (Fig. 4), alignment of only three different orthogonal twofold axes with point-group $D_2$ (Fig. 5) and no alignment of rotational axes (thus $C_1$)—we ask how the shells in **SD/Ag90a** align. The icosahedral outer shell of **SD/Ag90a** has fivefold axes, but these are absent in the octahedral middle and inner shells (Fig. 2b). The octahedral inner and middle shells

compatibly align all of their four-, three- and twofold axes, as in Fig. 3b. The icosahedral outer shell aligns four of its threefold axes with all four of the threefold axes of the octahedral shells (Fig. 6a) and three of its twofold axes with all three fourfold axes of the octahedral shells (Fig. 6b). Of note, the interstitial anions of $S^{2-}$ and $PO_4^{3-}$ also have important influences on aligning the three shells. Specifically, the threefold axes of the $Ag_6$, $Ag_{24}$, and $Ag_{60}$ shells pass through the $PO_4^{3-}$ ions, and the twofold axis of $Ag_{60}$ shell and fourfold axis of $Ag_{24}$ shell pass through the $S^{2-}$ ions, thus dictating the alignment of three different shells in the unique fashion discussed above. The alignment of these approximate polyhedra is nearly as good as in the same nest with regular polyhedra (Fig. 6c, d). Thus, the arrangement of the icosahedral and octahedral shells in the $Ag_{90}$ triple nest is the same as in Kepler's Kosmos (Fig. 4).

However, without a tetrahedral shell, **SD/Ag90a** is a subset of Kepler's Kosmos, with just $I_h$ (3.4.5.4) and $O_h$ (4.6.6 and 3.3.3.3) shells. As both $I_h$ and $O_h$ structures have inversion symmetry, their combination in **SD/Ag90a** also has inversion symmetry. Thus, the regular version of **SD/Ag90a** (Fig. 6c, d), with four threefold axes, three twofold axes, mirrors, inversion, and a symmetry order of 24, has $T_h$ symmetry (Supplementary Table 4).

**Optical properties of SD/Ag90a.** The solid-state ultraviolet–visible (UV/Vis) absorption spectra of **SD/Ag90a** and $[^tBuSAg]_n$ were measured at room temperature. As shown in Fig. 7a, **SD/Ag90a** exhibits a wide absorption range spanning UV and Vis regions with an absorption maximum at 419 nm. Compared with the absorption of $[^tBuSAg]_n$ at 280 nm, the absorption edge is

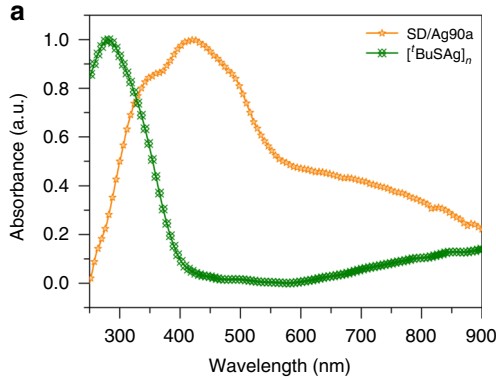

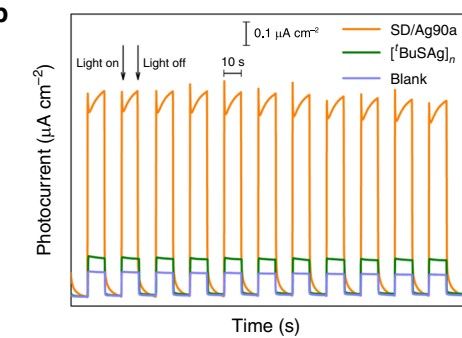

**Fig. 7 The UV–Vis spectra and photocurrent responses of SD/Ag90a and [$^t$BuSAg]$_n$. a** The normalized UV–Vis spectra of **SD/Ag90a** and [$^t$BuSAg]$_n$ precursor in the solid state. **b** Compared photocurrent responses of blank, **SD/Ag90a**, and [$^t$BuSAg]$_n$ ITO electrodes in a 0.2 M Na$_2$SO$_4$ aqueous solution under repetitive irradiation.

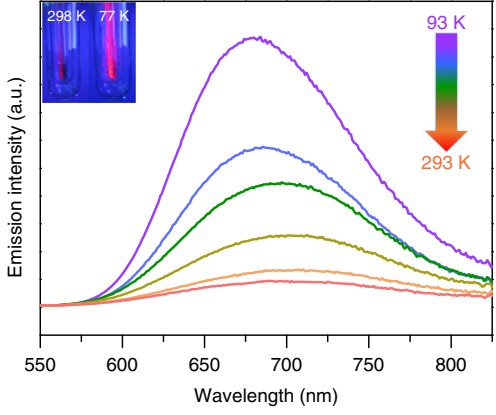

**Fig. 8 Varied-temperature luminescence spectra of SD/Ag90a from 93 to 293 K with 40 K as an interval.** Insets show the photographs of a sample of **SD/Ag90a** under a hand-held UV lamp (365 nm) at 298 and 77 K.

significantly red-shifted in **SD/Ag90a**, which should be caused by the ligand-to-metal charge transfer (LMCT) or/and cluster-centered (CC) transitions. Based on the Kubelka–Munk function of $(\alpha h\upsilon)^{1/2} = \kappa(h\upsilon - E_g)$ ($E_g$ is the band gap (eV), $h$ is Planck's constant (J·s), $\upsilon$ is the light frequency (s$^{-1}$), $\kappa$ is the absorption constant, and $\alpha$ is the absorption coefficient)[45], the band gaps of **SD/Ag90a** and [$^t$BuSAg]$_n$ precursor were determined as 0.69 and 2.34 eV, respectively (Supplementary Fig. 3), which indicates that the aggregation of multiple silver atoms into a cluster structure has an important influence on the HOMO–LUMO gap.

We further studied the photocurrent responses of **SD/Ag90a** and [$^t$BuSAg]$_n$ driven by visible-light in a typical three-electrode system (ITO glass as the working electrode, platinum wire as the assisting electrode and Ag/AgCl as the reference electrode) and keeping the bias voltage at 0.6 V. Upon on–off cycling irradiation with LED light ($\lambda = 420$ nm; 50 W; intervals of 10 s), clear photocurrent responses were observed for **SD/Ag90a** and [$^t$BuSAg]$_n$ (Fig. 7b). The photocurrent density of **SD/Ag90a** (0.9 μA cm$^{-2}$) was five times larger than that of [$^t$BuSAg]$_n$ (0.17 μA cm$^{-2}$), indicating that **SD/Ag90a** has better generation and separation efficiency of photoinduced electrons/holes pairs in ITO electrodes[46]. The generation of photocurrent may involve photoinduced charge migration from S 3$p$ to the Ag 5$s$ orbits.

The solid state emission spectra of **SD/Ag90a** were recorded as a function of temperature from 293 to 93 K with 40 K as an interval, showing luminescence thermochromic behavior (Fig. 8). The luminescence of **SD/Ag90a** originates from the ligand-to-metal charge transfer transition with a charge transfer from S 3$p$ to Ag 5$s$ orbitals and/or mixed with a cluster-centered transition related to Ag···Ag interactions[47]. **SD/Ag90a** shows gradually blue-shifted emissions from 700 to 684 nm ($\lambda_{ex} = 468$ nm) upon cooling, possibly related to enhanced molecular rigidity at lower temperatures. During the cooling process, the emission intensity shows a nearly tenfold increase from 293 to 93 K, which is attributed to reduction of the nonradiative decay at low temperature[48–50]. The emission lifetime of **SD/Ag90a** (Supplementary Fig. 4), falling in the microsecond scale ($\tau_{SD/Ag90a} = 17.80$ μs) at 93 K, suggests a triplet phosphorescence origin.

## Discussion

We have successfully synthesized an Ag$_{90}$ nanocluster with overall *pseudo*-T$_h$ symmetry. This silver nanocluster is divided into three shells as Ag$_6$@Ag$_{24}$@Ag$_{60}$ from inner to outer. The Ag$_6$ inner shell is an octahedron (a Platonic solid with 6 3.3.3.3 vertices), the Ag$_{24}$ middle shell is a truncated octahedron (an Archimedean solid with 24 4.6.6 vertices), and both have octahedral (O$_h$) symmetry. However, the Ag$_{60}$ outer shell is a rhombicosidodecahedron (an Archimedean solid with 60 3.4.5.4 vertices and icosahedral (I$_h$) symmetry). The **SD/Ag90a** nanocluster solves the apparent incompatibility among their symmetry groups with the most symmetric arrangement of two- and threefold axes. Creation of a nest with all three of the polyhedral symmetries, icosahedral, octahedral and tetrahedral—and resembling Kepler's Kosmos even more closely—remains an exciting challenge.

## Methods

**Syntheses of SD/Ag90a and SD/Ag90b**. [$^t$BuSAg]$_n$ (9.9 mg, 0.05 mmol), PhCOOAg (11.5 mg, 0.05 mmol), Na$_3$PO$_4$·12H$_2$O (5 mg, 0.013 mmol), and PhPO$_3$H$_2$ (4.7 mg, 0.03 mmol) were mixed in 6.5 mL MeOH/MeCN/DMF (v:v:v = 6:6:1). The resulting suspension was sealed and heated at 65 °C for 2000 min. After cooling to room temperature, dark brown crystals of **SD/Ag90a** were formed (yield: 30%). Combustion elementary analysis calculated (experimental) for **SD/Ag90a**: (C$_{204}$H$_{312}$Ag$_{90}$O$_{86}$P$_{26}$S$_{30}$): C, 15.69 (15.71%); H, 2.01 (1.99%). Selected IR peaks (cm$^{-1}$) of **SD/Ag90a**: 3664 (w), 2983 (w), 2894 (w), 1150 (w), 1104 (w), 1040 (m), 1007 (m), 928 (s), 747 (m), 715 (w), 689 (m), and 553 (s).

**SD/Ag90b** was synthesized similarly to **SD/Ag90a** but using CH$_3$SO$_3$Ag (10.2 mg, 0.05 mmol) instead of PhCOOAg. **SD/Ag90b** precipitated as red crystals from the evaporation of filtrate after solvothermal reaction for 1–2 weeks (yield: 19%).

## Data availability

The X-ray crystallographic coordinates for structures reported in this article have been deposited at the Cambridge Crystallographic Data Centre, under deposition numbers CCDC: 1913186–1913187. These data can be obtained free of charge from the Cambridge Crystallographic Data Centre via www.ccdc.cam.ac.uk/data_request/cif.

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

## Acknowledgements
This work was financially supported by the National Natural Science Foundation of China (Grant Nos. 91961105, 21822107, and 21571115), the Natural Science Foundation of Shandong Province (Nos. ZR2019ZD45, JQ201803, and ZR2017MB061), the Taishan Scholar Project of Shandong Province of China (Nos. tsqn201812003 and ts20190908), the Qilu Youth Scholar Funding of Shandong University, Project for Scientific Research Innovation Team of Young Scholar in Colleges, and Universities of Shandong Province (2019KJC028).

## Author contributions
D.S. conceived and designed the experiments; Y.-M.S and Z.W. conducted synthesis and characterization; Y.-M.S., Z.W., and D.S. performed research and analyzed data; C.-H.T. contributed to scientific discussion; S.S. analyzed the mathematics of the nested poly-hedral structure; D.S. and S.S. wrote the paper. All authors discussed the results and commented on the manuscript.

## Competing interests
The authors declare no competing interests.
