## [Peer Review File · Nature Communications]

Reviewers' comments:

Reviewer #1 (Remarks to the Author):

In this manuscript, the authors reported the synthesis and structure of a novel atomically precise Ag₉₀ nanocluster that consists of three well defined concentric shells, each with high symmetry, including an octahedral Ag₆ core, a middle Ag₂₄ shell with truncated octahedron shape, and an outer Ag₆₀ rhombicosidodecahedron shell. Another nontrivial feature of this Ag₉₀ cluster is that the three highly symmetric shells are aligned around multiple symmetry elements due to the presence of S₂- and PO₄₃- templates. Overall, this study provides important insight into the design of highly symmetric and complex atomically precise clusters via ligand template strategy, and will attract a broad interest from inorganic chemists, materials scientists, and in general those who are interested in structure design of crystalline materials. I recommend that the manuscript be accepted after a minor revision that addresses the following concerns:

1. In Figure 1 it is very difficult to appreciate the morphology of Ag₉₀b crystals from the microscope image. If possible, please make the images larger or have image with larger magnification. Scale bars are also missing.

2. In the last paragraph of page 3, it is stated that "Ag₆ octahedron with six 3.3.3 vertices ..." I think it should be "3.3.3.3 vertices".

3. Where is the mirror plane in the D_{2h} arrangement of the five platonic solids. I can only see the three orthogonal 2-fold axes. Please make sure if it has mirror planes or not, and point out the mirror plane explicitly. If no mirror planes are present, the symmetry would be D₂.

4. When discussing how the three shells are aligned around rotation axes in Ag₉₀, would it be meaningful to discuss the positions of S₂- and PO₄₃- templates again? I think they play an important role in aligning the three shells. For example, PO₄₃- ions seems to help align the 3-fold axes of the Ag₆, Ag₂₄ and Ag₆₀. And S₂- also seems to align C₂ of Ag₆₀ with C₄ of Ag₂₄.

Reviewer #2 (Remarks to the Author):

This paper from Sun group reports quite interesting results about the synthesis, geometry, and properties of atomically precise Keplerian Ag₉₀ nanoclusters that possess a three-shell nested metallic framework with each showing very charming aesthetic appreciation value. The cluster was synthesized by a unique anion template combination method (S₂- and PO₄₃-) to control the formation and separation of the different polyhedral shells of nanocluster, which gives very clean interfaces between different shells. More interesting, authors viewed these three polyhedral shells in a brand-new perspective which is related to the symmetry compatibility of polyhedra when posing them together.

They found three polyhedra with apparent incompatibility also can be engaged one-by-one together and explained such an uncommon mathematic issue using a product of chemistry as model through the geometrics method. Excepting for these fundamental science aspects, another noteworthy point of this work is the positive photocurrent responses by on-off cycling irradiation with LED light, which guarantees the possible applications of these clusters in the future. I thus believe that this manuscript shows important advances in the fields of chemistry, materials and mathematics and is suitable for publishing in such a prestigious comprehensive journal----Nature Comm after minor revision. Thus, I would like the authors to address these points shown below.

(1)Some names of abbreviated journals are wrong such as ref. 31 and 32, please carefully check and revise them.

(2)Authors have missed the page numbers for refs. 16 and 35.

(3)The proposed mixed anions strategy is fascinating. This method requires two different anions as templates. It would be good for the authors to provide a short comment on the underlying chemistry about why selecting PO₄³⁻ in this system.

(4)What is the special roles of S²⁻ and PO₄³⁻ on shaping the polyhedral silver shell? Some comments about this should be added into main text, which will be help for us to understand some intrinsic assembly rule in silver cluster chemistry.

(5)The innermost Ag₆ octahedron has been widely observed in silver cluster family for example in J. Am. Chem. Soc. 2019, 141, 19550 and Chem. Commun., 2013, 49, 376. Is this one also subvalent as reported by the same group? How to determine the valence of the inner core?

(6)In synthesis section, no mol amount was provided for Na₃PO₄, why?

Reviewer #3 (Remarks to the Author):

In this paper, Sun and Schein performed a detailed and in-depth investigations on a pair of polymorphic Ag₉₀ nanoclusters under the direction of mixed anion templates, S²⁻ and PO₄³⁻. It is thrilled to find that the as-synthesized Ag₉₀ nanoclusters have three polyhedral shells from inner to outer, octahedron, truncated octahedron, and rhombicosidodecahedron, respectively. The former two have Oh symmetry whereas the latter has Ih symmetry. The nesting of three polyhedra in one compound thus arouses an interesting mathematic problem that is how they can align together by fitting some unmatched symmetry elements such as 2-, 3-, or 4-fold axis. This problem was well explained in this work based on geometric principles. The manuscript is well written with solid data to support their conclusions. The quality of single crystal x-ray data was quite good in this work, although I know the huge difficulty for collecting satisfactory crystallographic data for structural analysis of such large nanoclusters. Moreover, temperature-dependent emission and photocurrent response properties were also studied, providing an insight of the self-assembly and demonstrating the potential application of such high-nuclearity clusters. I am sure that the synthetic protocol, structural aspects and new findings in math included in this work will be of interest to heterogeneous readers from communities of coordination chemistry, nanocluster chemistry, materials science, nanoscience, and even mathematics. I should not hesitate to recommend the acceptance of this paper in Nature Comm but after the authors have addressed the following minor

issues.

1. Why PhCOOAg and CH₃SO₃Ag can produce two Ag₉₀ nanoclusters packing in the different unit cell? How about the synthesis results if other silver salts used instead? Some comments about this will be welcomed in main text.
2. Are these silver nanoclusters stable in air or light?
3. The discussion of optical properties of the title compound raises concerns. They apply the Kubelka-Munk function to calculate the optical band gap as if they had a band semiconductor. But the authors did not show which type of the function used, which is important because different functions are used for direct-gap and indirect-gap semiconductors.
4. In this work, I found that single-crystal X-ray diffraction was used as a main technique to resolve/refine the structure and determine the molecular composition of the as-synthesized Ag₉₀ nanoclusters. However, it has been already well accepted in the nanocluster field that ESI-MS is an alternative powerful method to determine the molecular formula, charge state, solution behavior, cluster transformation reaction, even the assembly mechanism of the metal nanoclusters. What is the ESI-MS result of Ag₉₀ nanoclusters in the current work? Some possible discussions on ESI-MS may benefit the readers, especially to those lack of extensive research experiences on metal nanoclusters.
5. As stated by authors, all S²⁻ ions are from in situ decomposition of tBuSH. I think there should have some references to support this hypothesis.
6. The text needs polishing or proof-reading due to the existence of several typos.

Reviewer #1 (Remarks to the Author):

In this manuscript, the authors reported the synthesis and structure of a novel atomically precise Ag₉₀ nanocluster that consists of three well defined concentric shells, each with high symmetry, including an octahedral Ag₆ core, a middle Ag₂₄ shell with truncated octahedron shape, and an outer Ag₆₀ rhombicosidodecahedron shell. Another nontrivial feature of this Ag₉₀ cluster is that the three highly symmetric shells are aligned around multiple symmetry elements due to the presence of S²⁻ and PO₄³⁻ templates. Overall, this study provides important insight into the design of highly symmetric and complex atomically precise clusters via ligand template strategy, and will attract a broad interest from inorganic chemists, materials scientists, and in general those who are interested in structure design of crystalline materials . I recommend that the manuscript be accepted after a minor revision that addresses the following concerns:

Response: We are very pleased and excited by positive comments on the novelty and significance of this work. We also believe that, thanks to your comments, the quality of the manuscript has been improved.

1. In Figure 1 it is very difficult to appreciate the morphology of Ag90b crystals from the microscope image. If possible, please make the images larger or have image with larger magnification. Scale bars are also missing.

Response: Thank you for your constructive suggestion/reminder. We have enlarged the images of crystals in Figure 1 and put in scale bars for them (see below). The figure caption was also revised correspondingly.

2. In the last paragraph of page 3, it is stated that “Ag₆ octahedron with six 3.3.3 vertices ...” I think it should be “3.3.3.3 vertices”.

Response: Thanks for your careful checking. You are perfectly correct. We have now made the correction.

3. Where is the mirror plane in the D_{2h} arrangement of the five platonic solids. I can only see the three orthogonal 2-fold axes. Please make sure if it has mirror planes or not, and point out the mirror plane explicitly. If no mirror planes are present, the symmetry would be D₂.

Response: Wow! We are so grateful to the reviewer for catching this error. We have revised the symmetry group to D₂ in the figure legend, the text and the table.

(4) When discussing how the three shells are aligned around rotation axes in Ag₉₀, would it be meaningful to discuss the positions of S²⁻ and PO₄³⁻ templates again? I think they play an important role in aligning the three shells. For example, PO₄³⁻ ions seems to help align the 3-fold axes of the Ag₆, Ag₂₄ and Ag₆₀. And S²⁻ also seems to align C₂ of Ag₆₀ with C₄ of Ag₂₄.

Response: Thanks for your constructive suggestions. Yes, the positions of S²⁻ and PO₄³⁻ templates are important for aligning the three shell by templation effect. Roughly, 3-fold axes of the Ag₆, Ag₂₄ and Ag₆₀ shells pass through the PO₄³⁻ ions, whereas S²⁻ ions are passed through 2-fold axes of the Ag₆₀ shell and 4-fold axes of the Ag₂₄ shell, all of which also dictate the alignment of three different shells in such a special fashion as discussed in main text. These comments were also added into main text as following: “Of note, the interstitial anions of S²⁻ and PO₄³⁻ also have important influences on aligning the three shells. Specifically, the 3-fold axes of the Ag₆, Ag₂₄ and Ag₆₀ shells pass through the PO₄³⁻ ions, and the 2-fold axis of Ag₆₀ shell and 4-fold axis of Ag₂₄ shell pass through the S²⁻ ions, thus dictating the alignment of three different shells in the unique fashion discussed above.”.

Reviewer #2 (Remarks to the Author):

This paper from Sun group reports quite interesting results about the synthesis, geometry, and properties of atomically precise Keplerian Ag₉₀ nanoclusters that possess a three-shell nested metallic framework with each showing very charming aesthetic appreciation value. The cluster was synthesized by a unique anion template combination method (S²⁻ and PO₄³⁻) to control the formation and separation of the different polyhedral shells of nanocluster, which gives very clean interfaces between different shells. More interesting, authors viewed these three polyhedral shells in a brand-new perspective which is related to the symmetry compatibility of polyhedra when posing them together. They found three polyhedra with apparent incompatibility also can be engaged one-by-one together and explained such an uncommon mathematic issue using a product of chemistry as model through the geometrics method. Excepting for these fundamental science aspects, another noteworthy point of this work is the positive photocurrent responses by on-off cycling irradiation with LED light, which guarantees the possible applications of these clusters in the future. I thus believe that this manuscript shows important advances in the fields of chemistry, materials and mathematics and is suitable for publishing in such a prestigious comprehensive journal----*Nature Comm* after minor revision. Thus, I would like the authors to address these points shown below.

Response: Thank you for your positive comments on our work in the synthesis of the large metal clusters that possess a three-shell nested metallic framework with apparent symmetry incompatibility. We would also like to thank the reviewer for his/her inspiring and constructive comments/suggestions, which have been taken into careful consideration in this revision. We believe that the quality of the manuscript has been improved thanks to these constructive suggestions.

(1) Some names of abbreviated journals are wrong such as ref. 31 and 32, please carefully check and revise them.

Response: Thanks for your careful checking. We have carefully checked abbreviated names of all references and corrected these errors.

(2) Authors have missed the page numbers for refs. 16 and 35.

Response: Thank you again for your careful checking. We have carefully checked the page number of all references and corrected these errors. Please note, there is no end page number available for papers published in *Nature Commun.*

(3) The proposed mixed anions strategy is fascinating. This method requires two different anions as templates. It would be good for the authors to provide a short comment on the underlying chemistry about why selecting PO_4^{3-} in this system.

Response: Thank you for your constructive suggestions. Yes, anion template is really powerful in directing the syntheses of large silver clusters. Here we stressed the importance of size, charge state, and geometry of mixed anions in the formation of interesting 90-nuclei silver clusters. Previously, these anions, such as S^{2-} (*Nanoscale* 9, (2017), 3601-3608), halogen anions (Cl^- , Br^- , I^-) (*Eur. J. Inorg. Chem.* (2010), 2084-2087; *Chem. Commun.* (2008)5586-5588; *J. Am. Chem. Soc.* 134, (2012), 2922-2925), small oxo-anions (NO_3^- , CO_3^{2-} , ClO_4^- , SO_4^{2-} , CrO_4^{2-} , MoO_4^{2-}) (*Chem. Eur. J.* 22, (2016), 6830-6836; *Nanoscale* 9, (2017), 5305-5314; *Nature Commun.* 9, 2094, (2018); *Chem. Commun.* 54, (2018), 2361-2364), large polyoxometallates ($\text{W}_6\text{O}_{21}^{6-}$, $\text{Mo}_{20}\text{O}_{66}^{12-}$, $\text{V}_{10}^{\text{V}}\text{V}_{2}^{\text{IV}}\text{O}_{34}^{10-}$) (*Chem. Commun.* 48, (2012), 5844-5846; *Nanoscale* 7, (2015), 7151-7154; *Angew. Chem. Int. Ed.* 55, (2016), 3699-3703), have been used in synthesizing diverse silver clusters. Selective intercalation of the hetero-anions templates may make us better explore the template behavior of different anions on shaping silver polygons, then silver polyhedra. Based on these considerations, PO_4^{3-} and S^{2-} in one cluster have not been explored, even only PO_4^{3-} templated silver nanocluster is also absent. PO_4^{3-} has higher negative charge compared to other tetrahedral anions such as SO_4^{2-} and ClO_4^- , which can aggregate more Ag^+ ions to form high-nuclearity clusters through electrostatic attraction. Moreover, the T_d symmetry of PO_4^{3-} ion also can shape high-symmetry clusters as seen in Ag_{90} clusters. These comments were added into main text as following: “The PO_4^{3-} ion was considered as anion template mainly due to its high negative charge and T_d symmetry. The former feature can aggregate more Ag^+ ions to form high-nuclearity cluster through electrostatic

attraction, whereas the latter can shape cluster with a T-related symmetry”.

(4) What is the special roles of S^{2-} and PO_4^{3-} on shaping the polyhedral silver shell?

Some comments about this should be added into main text, which will be help for us to understand some intrinsic assembly rule in silver cluster chemistry.

Response: Thank you for this constructive suggestion as well. Based on the structure analysis for Ag_{90} clusters, we found that the PO_4^{3-} ion has some role in shaping silver 3-gons and 6-gons which are essential elements to construct icosahedral the Ag_{60} rhombicosidodecahedron and the octahedral Ag_{24} truncated octahedron, respectively. The spherical S^{2-} ion helps to fabricate the silver tetragon, which is an essential element for both the rhombicosidodecahedron and the truncated octahedron. These discussions were also added into main text as following: “Based on the above analysis, we found that the tetrahedral PO_4^{3-} ion has a special role in shaping silver 3-gons and 6-gons, essential elements to construct the rhombicosidodecahedron and the truncated octahedron, respectively. As for the spherical S^{2-} ion, it assists in fabricating the silver 4-gon, an essential element for both the rhombicosidodecahedron and the truncated octahedron. Both inorganic anions act not only as templates to shape the silver polyhedra by defining the essential polygon elements but also function as glue to consolidate the overall nested silver shells.”.

(5) The innermost Ag_6 octahedron has been widely observed in silver cluster family for example in *J. Am. Chem. Soc.* 2019, 141, 19550 and *Chem. Commun.*, 2013, 49, 376. Is this one also subvalent as reported by the same group? How to determine the valence of the inner core?

Response: Thank you yet again for another constructive suggestion. The innermost Ag_6 octahedron is constructed by pure $Ag(I)$ ions without any subvalent feature. This conclusion is very clear because the quite long $Ag\cdots Ag$ edge lengths (3.51-3.61 Å), which are longer than twice the Van der Waals radius of silver(I) ion (3.44 Å), cannot be deemed as argentophilic interactions. We also reported several cases containing a subvalent Ag_6 octahedron in silver clusters (see *Nature Comm.*, 9, (2018) 2094; *Chem. Commun.*, 55, (2019) 10296; *Chem.*

Commun., 47(2011) 1461; *Sci China Chem*, 63, (2020) 16), in which we found that Ag...Ag distances in subvalent Ag₆ octahedron are generally short and locate around 2.88 Å. This criterion helps us to rule out the existence of a subvalent nature for the Ag atoms in the Ag₆ octahedron in Ag₉₀ clusters. Moreover, the whole molecular electroneutrality consideration also supports the conclusion that all silver atoms have a +1 oxidation state [$90 (\text{Ag}^+) = 24 (8 \times \text{PO}_4^{3-}) + 12 (6 \times \text{S}^{2-}) + 24 (24 \times \text{tBuS}^-) + 24 (12 \times \text{PhPO}_3^{2-}) + 6 (6 \times \text{PhPO}_3\text{H}^-)$]. We added these discussions into main text along with some suitable references as following: “Of note, the long Ag...Ag edges in the Ag₆ octahedron also rule out a subvalent nature, which usually produces short Ag...Ag distances approximating to 2.88 Å.³⁹”.

(6) In synthesis section, no mol amount was provided for Na₃PO₄, why?

Response: Thank you again for carefully checking. We apologize for omitting the mol amount for Na₃PO₄ in the synthesis section. During this revision stage, we added this important information as “Na₃PO₄·12H₂O (5 mg, 0.013 mmol)”.

Reviewer #3 (Remarks to the Author):

In this paper, Sun and Schein performed a detailed and in-depth investigations on a pair of polymorphic Ag₉₀ nanoclusters under the direction of mixed anion templates, S²⁻ and PO₄³⁻. It is thrilled to find that the as-synthesized Ag₉₀ nanoclusters have three polyhedral shells from inner to outer, octahedron, truncated octahedron, and rhombicosidodecahedron, respectively. The former two have O_h symmetry whereas the latter has I_h symmetry. The nesting of three polyhedra in one compound thus arouses an interesting mathematic problem that is how they can align together by fitting some unmatched symmetry elements such as 2-, 3-, or 4-fold axis. This problem was well explained in this work based on geometric principles. The manuscript is well written with solid data to support their conclusions. The quality of single crystal x-ray data was quite good in this work, although I know the huge difficulty for collecting satisfactory crystallographic data for structural analysis of such large nanoclusters. Moreover, temperature-dependent emission and photocurrent response properties were also studied, providing an insight of the self-assembly and demonstrating the potential application of such high-nuclearity clusters. I am sure that the synthetic protocol, structural aspects and new findings in math included in this work will be of interest to heterogeneous readers from communities of coordination chemistry, nanocluster chemistry, materials science, nanoscience, and even mathematics. I should not hesitate to recommend the acceptance of this paper in *Nature Comm* but after the authors have addressed the following minor issues.

Response: We are pleased and excited by the reviewer's positive assessment of the novelty and significance of our study. We would also like to thank the reviewer for his/her inspiring and constructive comments/suggestions, which have been taken into careful consideration in this revision. We believe that the revised manuscript is an improvement thanks to your constructive suggestions.

1. Why PhCOOAg and CH₃SO₃Ag can produce two Ag₉₀ nanoclusters packing in the different unit cell? How about the synthesis results if other silver salts used instead? Some comments about this will be welcomed in main text.

Response: Thank you for your constructive suggestion. Although these different anions, PhCOO^- and CH_3SO_3^- , did not participate in the final structures of the Ag_{90} clusters, they may influence the crystallization process of these clusters through supramolecular interactions such as hydrogen bonds, which determines the ultimate crystalline phases. Other common silver salts were tried in the above synthesis experiments, but none produced desired clusters. We also added the following discussions into the main text: “Although the different anions, PhCOO^- and CH_3SO_3^- , did not participate in the final structures of the Ag_{90} clusters, they may have influenced the crystallization process through supramolecular interactions such as hydrogen bonds, which causes the formation of the ultimate crystalline phases. Other common silver salts, such as AgBF_4 , CF_3COOAg , $\text{CF}_3\text{SO}_3\text{Ag}$, and AgNO_3 , were tried in the above synthesis experiments, but none produced desired clusters.”.

2. Are these silver nanoclusters stable in air or light?

Response: Interesting question. The Ag_{90} nanoclusters are quite stable because the bulk sample of them can keep the color and morphology unchanged for at least one month under ambient conditions, which suggests they are stable to light, atmospheric moisture and oxygen in the solid state. We added this comment into main text as following: “They are stable under ambient conditions because the bulk sample of them can keep the color and morphology unchanged for at least one month.”

3. The discussion of optical properties of the title compound raises concerns. They apply the Kubelka-Munk function to calculate the optical band gap as if they had a band semiconductor. But the authors did not show which type of the function used, which is important because different functions are used for direct-gap and indirect-gap semiconductors.

Response: Thank for this constructive suggestion. We used the function of $(\alpha h\nu)^{1/2} = \kappa(h\nu - E_g)$ (*Catal Today* 250, (2015), 95–101; *J Sol-Gel Sci Techn* 61, (2012), 1-7 and *ACS Nano* 4, (2010), 5813-5818.) to calculate the optical band gap. Here, E_g is the band gap (eV), h is the Planck's constant (J.s), ν is the light frequency (s^{-1}), κ is the absorption constant and α is the absorption coefficient.

We added the Kubelka-Munk function in the main text as follows: ”Based on the Kubelka-Munk function of $(\alpha hv)^{1/2} = \kappa(hv-E_g)$ (E_g is the band gap (eV), h is the Planck’s constant (J.s), v is the light frequency (s^{-1}), κ is the absorption constant and α is the absorption coefficient),” **The related reference was also added as ref. 45.**

4. In this work, I found that single-crystal X-ray diffraction was used as a main technique to resolve/refine the structure and determine the molecular composition of the as-synthesized Ag₉₀ nanoclusters. However, it has been already well accepted in the nanocluster field that ESI-MS is an alternative powerful method to determine the molecular formula, charge state, solution behavior, cluster transformation reaction, even the assembly mechanism of the metal nanoclusters. What is the ESI-MS result of Ag₉₀ nanoclusters in the current work? Some possible discussions on ESI-MS may benefit the readers, especially to those lack of extensive research experiences on metal nanoclusters.

Response: Thank you for this suggestion. Yes, electrospray ionization mass spectrometry (ESI-MS) has become a powerful tool in cluster chemistry to investigate the composition, charge state, solution behavior, cluster transformation reaction, assembly mechanism and its inverse process, its dissociation pathway in gas phase. This technique has also been widely used in our laboratory, and several reports from our group have verified the advantages of it (see: *Proc Natl Acad Sci USA* 114, (2017) 12132; *J. Am. Chem. Soc.*, 138, (2016) 1328; *J. Am. Chem. Soc.*, 139, (2017) 14033; *Nature Comm.*, 9, (2018) 2094; *Nature Comm.*, 9, (2018) 4407; *Nature Comm.*, 11, (2020) 308). The successful collection of a mass spectrum is closely related to the parameters of the instruments and the intrinsic properties of the compound. In this work, we really tried to investigate the solution behaviors of Ag₉₀ cluster by using mass spectrometers of different models (Agilent 6224 ESI-TOF and Bruker impact II ESI-TOF) under different instrumental parameters; however, to our disappointment, we did not get any useful/informative ESI-MS signals in both positive and negative-ion modes for the two clusters. The reason may be the compounds themselves. For example, the neutral Ag₉₀ nanocluster may be hard

to ionize under the current mass spectrometer conditions, even when we added CsOAc to help ionization. Also, the nanoclusters are extremely unstable and may be badly fragmented in the ESI-MS process. We added a short comment about the ESI-MS result in the main text as following: “The electrospray ionization mass spectrometry (ESI-MS) of SD/Ag90a dissolved in CH₂Cl₂ or CH₃OH did not give useful valuable data, which indicates that either i) SD/Ag90a is fragmented during the ionization process or ii) it is neutral and is hard to ionize under mass spectrometer conditions – even when we added CsOAc to aid in ionization.³⁸”. The related reference was also added as ref. 38.

5. As stated by authors, all S²⁻ ions are from in situ decomposition of ^tBuSH. I think there should have some references to support this hypothesis.

Response: Thank you for for this suggestion. We have reference 28 and this text: “Six μ₈-S²⁻ ions from in situ decomposition of ^tBuSH intercalate the aperture between Ag₂₄ and Ag₆₀ shells (Ag-S: 2.43-2.89 Å) by linking two 4-gons up and down from these shells, respectively (Fig. 2h).²⁸”

6. The text needs polishing or proof-reading due to the existence of several typos.

Response: Thank you for finding these. We and other co-authors have carefully checked the overall manuscript and removed the typos.

REVIEWERS' COMMENTS:

Reviewer #2 (Remarks to the Author):

I carefully read the revised paper. The authors have properly addressed the reviewers' comments. I think this manuscript can be accepted in Nature Comm.

Reviewer #3 (Remarks to the Author):

The authors have addressed all my concerns. I would like to recommend the acceptance.

RESPONSE TO REVIEWERS' COMMENTS:

Reviewer #2 (Remarks to the Author):

I carefully read the revised paper. The authors have properly addressed the reviewers' comments. I think this manuscript can be accepted in Nature Comm.

Response: Thank you very much. We appreciate your recommendation for publication of our work in Nature Communications.

Reviewer #3 (Remarks to the Author):

The authors have addressed all my concerns. I would like to recommend the acceptance.

Response: Thank you very much. We appreciate your recommendation for publication of our work in Nature Communications.